# Fatal Bronchopneumonia and Tracheitis in a Green Turtle (*Chelonia mydas*) Caused by *Serratia proteamaculans*

**DOI:** 10.3390/ani12151891

**Published:** 2022-07-25

**Authors:** Jane Hall, Hannah Bender, Natalie Miller, Paul Thompson

**Affiliations:** 1Australian Registry of Wildlife Health, Taronga Conservation Society Australia, Bradleys Head Road, Mosman, NSW 2088, Australia; hanbender@gmail.com; 2Taronga Wildlife Hospital, Taronga Conservation Society Australia, Bradleys Head Road, Mosman, NSW 2088, Australia; nmiller@zoo.nsw.gov.au (N.M.); pthompson@zoo.nsw.gov.au (P.T.)

**Keywords:** sea turtles, pathology, *Serratia proteamaculans*, tracheitis, pneumonia

## Abstract

**Simple Summary:**

Chelonian respiratory disease is often challenging to diagnose and treat due to unique species characteristics, environmental factors, co-morbidities, and limitations in diagnostic capabilities. Infections caused by previously unreported or poorly understood infectious agents may further complicate treatment efforts. Here, we describe an unusual case of fatal respiratory disease in a green turtle associated with *Serratia proteamaculans* infection. *S. proteamaculans* is a plant pathogen and possible cross-kingdom infectious agent, not previously reported to cause disease in reptiles. This case emphasizes the importance of microbial culture in managing chelonian respiratory disease and considers the potential for disease arising from the industrial or agricultural application of phytopathogens. In this report, we review the importance of understanding those bacteria that are often dismissed as opportunistic, explore treatment options for those times when opportunity leads to infection, and discuss the importance of bacterial culture and sensitivity in the respiratory disease of marine turtles. We also discuss the bacterium *Serratia proteamaculans*, a possible cross-kingdom pathogen, and invite the reader to think about the impact of biological alternatives to anthropogenic chemical use and the effects these solutions may have when environmental loads are artificially increased.

**Abstract:**

A free-ranging subadult, male green turtle (*Chelonia mydas*) presented with radiographic evidence of pneumonia and died acutely. On necropsy, the trachea and bronchi were plugged by diphtheritic membranes, comprised of fibrin, necrotic debris, and colonies of bacilli, identified as *Serratia proteamaculans*. *S. proteamaculans*, typically considered an opportunistic plant pathogen, has rarely been described as causing disease in animals. This is the first report of *S. proteamaculans* causing severe necrotizing tracheitis and bronchopneumonia in a reptile.

## 1. Introduction

Respiratory diseases are common in free-ranging wild chelonians [1]. Various bacterial and fungal pathogens have been cultured from pulmonary lesions in debilitated turtles, and most are described as opportunistic pathogens following a primary insult including traumatic injuries, such as a boat strike or foreign body ingestion, or a primary viral infection [1,2]. Environmental and climatic conditions have also been implicated as important drivers of immunosuppression and disease in chelonians, most notably in North America and Europe where, when water temperatures drop below 10 °C (50 °F), sea turtles may experience “cold stunning” [3,4]. Cold-stunning may present clinically as lethargy, shock, pneumonia, and death [3,4]. The present report documents an unusual case of fatal bronchopneumonia and tracheitis in a green turtle (*Chelonia mydas*) associated with colonization by the bacterium *Serratia proteamaculans*.

A subadult male green turtle was found floating in a weakened and poorly responsive state near the shoreline at Gymea Bay (34°2′59″ S 151°5′11″ E), New South Wales, Australia, in September 2020. Average water temperatures at this time were approximately 18 °C (64 °F). The turtle was transported to the Taronga Wildlife Hospital (TWH) at Taronga Zoo, Sydney, for a veterinary examination. Craniocaudal and dorsoventral radiographs revealed bilaterally extensive areas of increased pulmonary opacity (Figure 1A). Despite supportive care, the turtle was found dead two days later and a post-mortem examination was undertaken immediately to identify the cause of death.

## 2. Materials and Methods

On necropsy, representative samples from each organ were fixed in 10% neutral buffered formalin before routine histopathology processing in paraffin wax with hematoxylin and eosin staining. Tissues examined included trachea, lung, thyroid gland, colon, spleen, liver, pancreas, kidney, skeletal muscle, heart, testis, esophagus, small intestine, and brain. Sections of trachea and lung were additionally stained with Ziehl–Neelsen (ZN), Gram Twort, and Gomori’s methenamine silver (GMS) stains. 

Fresh samples of the kidney, liver, and bronchus were collected under sterile conditions and submitted for routine culture at the TWH clinical pathology laboratory. Tissue impression smears stained with Gram and ZN were examined under 100× oil immersion. Inoculated horse blood agar (HBA) and MacConkey agar (MAC) (Thermo Fisher Scientific, Scoresby, Victoria, Australia) were incubated at 35 °C in 4.5% carbon dioxide incubated for 24–48 h. Additional inoculated HBA was left at room temperature for 24–48 h. HBA anaerobic (ANA) agar (Thermo Fisher Scientific, Scoresby, Victoria, Australia) was incubated anaerobically for 24–48 h at 35 °C. A Sabouraud with antibiotics (SAB+) agar plate (Thermo Fisher Scientific, Scoresby, VIC, Australia) was also inoculated to look for fungal pathogens and incubated aerobically for 6 weeks at 25 °C.

Significant bacterial isolates were submitted to the University of Sydney, Veterinary Pathology Diagnostic Service (VPDS) for identification via matrix-assisted laser desorption/ionization time-of-flight (MALDI-ToF) mass spectrometry, utilizing the Bruker MALDI-ToF Biotyper system (Brucker, Billerica, MA, USA). 

Antibiotic susceptibility testing was performed on pure cultures of significant bacterial isolates in-house at the TWH clinical pathology laboratory. The Calibrated Dichotomous Susceptibility (CDS) method was employed [5]. Antibiotic impregnated discs tested included amoxicillin/clavulanic acid, ampicillin, cefotaxime, ceftazidime, cotrimoxazole, enrofloxacin, gentamicin, imipenem, piperacillin, and tazocin. An imipenem disc placed near the ceftazidime disc was also used to induce any potential AmpC β-lactamase production.

## 3. Results

### 3.1. Necropsy

The gross examination of the turtle, which weighed 15.8 kg (curved carapace length 515 mm), showed minimal autolysis and good body condition with moderate fat deposits and adequate muscle mass. Several large barnacles adhered to the carapace and plastron and there were few small barnacles adhered to the flippers. A small amount of red-tinged fluid exuded from the mouth. The coelomic cavity contained a large volume of red-tinged fluid and yellow fibrin strands adhered to the serosal surfaces. The glottis, trachea, and primary bronchi were partially occluded by a thick layer of tenacious white-tan material (Figure 1B–D), which extended into the right bronchial tree. The left bronchial mucosa was coated with caseous exudate, and the lumen contained blood-tinged, foamy fluid. Annular ligaments of the trachea appeared hemorrhagic (Figure 1B,C), as did the lung lobes (Figure 1D). 

### 3.2. Histopathology

Histologic sections of the trachea were characterized by extensive mucosal ulceration with replacements by thick, lamellated layers of fibrin admixed with abundant macrophages and frequent colonies of variably rod-shaped to elongated Gram-negative bacilli (Figure 2). The remnant mucosa was congested with fibrinoid necrosis of mucosal and submucosal blood vessels and submucosal edema (Figure 2). 

In sections of the lung, airways were occluded by fibrin and hemorrhage interspersed with florid colonies of small Gram-negative bacilli, and the pulmonary parenchyma was extensively effaced by necrosis and hemorrhage (Figure 2). No acid-fast bacteria or fungal elements were detected in the trachea or lungs with additional ZN and GMS stains, respectively. There were no other significant findings.

### 3.3. Microbiology

There was no evidence of acid-fast bacteria on ZN; however, the bronchus did show an occasional Gram-negative rod on the Gram stain. The liver showed no growth aerobically; no tissues showed anaerobic or fungal growth, or growth at room temperature, after the maximum incubation periods. After 24 h, both the incubated HBA and MAC agars grew pure, light growth of a white, oxidase negative coliform from the kidney and a pure, heavy growth of the same colony type from the bronchus. The coliform identified in the kidney and bronchus was subcultured onto a fresh HBA plate and incubated for a further 24 h at 35 °C in 4.5% carbon dioxide before being forwarded to VPDS for MALDI-ToF identification. The isolate was identified as *Serratia proteamaculans* with a score value of ≥2, indicating a high degree of accuracy. 

Antibiotic susceptibility testing indicated that this *S. proteamaculans* was resistant to amoxicillin/clavulanic acid, ampicillin, and cefotaxime. The bacterium was susceptible to ceftazidime, co-trimoxazole, enrofloxacin, gentamicin, piperacillin, tazocin, and imipenem. The presence of an inducible AmpC β-lactamase was not detected. 

## 4. Discussion

Necropsy findings in this turtle, consistent with radiographic changes, were most significant within the respiratory tract, with partial occlusion of large airways by tenacious diphtheritic membranes. Histologic changes within the trachea and lung were consistent with fibrinonecrotizing tracheitis and bronchopneumonia, respectively. Frequent bacterial colonies were present within the inflammatory exudate and aerobic culture isolated a pure growth of *Serratia proteamaculans* from samples of both the bronchus and kidney, suggestive of multisystemic infection. Histologic changes were consistent with an acute course of disease and no evidence of chronic underlying disease or debility was present during necropsy. Within sections of remnant tracheal mucosa and pulmonary parenchyma, there was no evidence of concurrent viral infection, and it is concluded that pneumonia and death were caused by infection with *S. proteamaculans*. 

*S. proteamaculans* is an opportunistic, aerobic, mesophilic pathogen in the *S. liquefaciens* complex. The organism is ubiquitous and has been isolated from soil, plants, water, and insects [6,7]. Nosocomial infections of *S. marcescens* have been reported in both human and veterinary hospital settings, generally associated with intubation, catheterization, or other invasive procedures [8,9]. *Serratia* spp. are recognized as important opportunistic pathogens and are routinely isolated from wounds, blood, and respiratory and urinary samples [8]. In 1993, Bollet et. al. reported the first isolation of *S. quinovora*, then described as a subspecies of *S. proteamaculans*, from a patient presenting with a suffocating oral abscess that progressed to bilateral pneumonia and renal failure [6]. 

*S. proteamaculans* is the only described phytopathogenic *Serratia* species causing leaf spot disease of *Protea cynaroides* [7]. *S. proteamaculans* is also an important bacterial pesticide causing amber disease in grass grub (*Costelytra giveni*) and manuka beetle (*Pyronota* sp.) larvae in New Zealand [10,11]. The phytopathogenic, biosurfactant, and hemolytic characteristics of *Serratia* spp. have given rise to various pharmaceutical, industrial, agricultural, and environmental applications with potential for bioremediation and agricultural and marine applications as they are more environmentally suitable than their synthetic counterparts [12,13,14]. While clinical phytopathogenic infections are generally considered opportunistic in animals, Kim et al. [15] highlight the importance of cross-kingdom pathogens and the mechanisms these organisms may employ to effectively and actively cross defense barriers. The *S. proteamaculans* cultured in this case was resistant to the β-lactam antibiotics amoxicillin/clavulanic acid, ampicillin, and cefotaxime but susceptible to ceftazidime, piperacillin, and tazocin. An inducible AmpC beta-lactamase was not detected by the phenotypic analysis. The isolate was also susceptible to enrofloxacin, a fluoroquinolone antibiotic, and the aminoglycoside antibiotic gentamicin. β-lactam antibiotic resistance has been reported for several *Serratia* spp., including *S. proteamaculans* [7] and chlorhexidine-resistant strains of *S. marcescens* in the veterinary setting [9]. This highlights the importance of microbial culture and antibiotic susceptibility testing prior to antibiotic therapy for all marine turtles entering veterinary and rehabilitation settings. The turtle in this case succumbed to its infection prior to any antibiotic therapy being administered. 

## 5. Conclusions

This is the first documented case of infection caused by *S. proteamaculans* in any reptilian species. Given the ubiquity of the organism, *S. proteamaculans* should be considered in chelonians presenting with acute exudative respiratory tract infections. Microbial culture and sensitivity testing should also be employed to ensure effective antibiotic therapy for the treatment of marine turtles in care.

## Figures and Tables

**Figure 1 animals-12-01891-f001:**
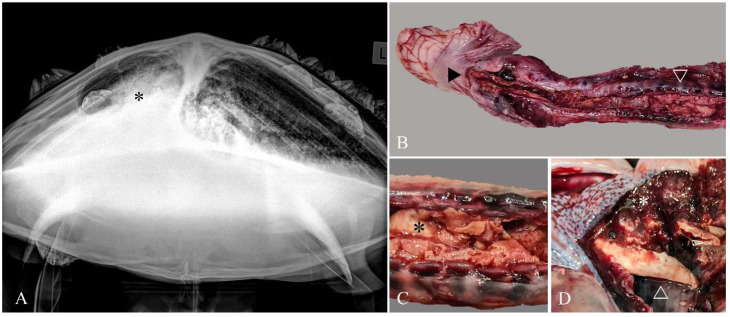
Necrotizing tracheitis and bronchopneumonia due to *Serratia proteamaculans* infection. Imaging and gross pathology. (**A**) Extensive areas of increased soft tissue opacity (*) were present bilaterally on craniocaudal radiographs. (**B**) From the glottis (►), the tracheal mucosa was extensively ulcerated (∇) and replaced by diphtheritic membranes composed of fibrin and necrotic debris. (**C**) Ulcerated tracheal mucosa filled with diphtheritic membrane (*), and (**D**) Bronchi partially occluded with a diphtheritic membrane plug (Δ) and hemorrhagic pulmonary parenchyma (*).

**Figure 2 animals-12-01891-f002:**
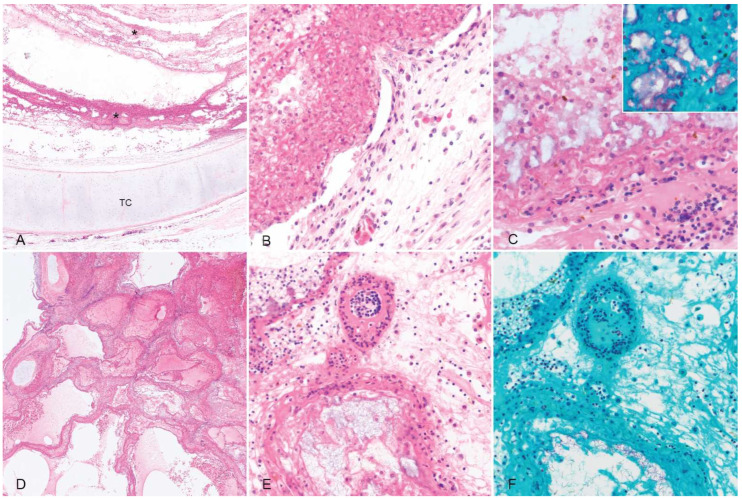
Histopathology. (**A**) Trachea. The mucosa is ulcerated and replaced by a diphtheritic membrane composed of lamellated layers of fibrin and necrotic cell debris (*). The denuded tracheal cartilage (TC) is located at the lower margin of the figure. Hematoxylin and eosin (HE). (**B**) Higher magnification image of the ulcerated tracheal mucosa. (**C**) Abundant elongated bacilli are scattered throughout the necrotic cell debris. Inset image: Bacteria are Gram-negative. Gram–Twort stain. (**D**) Lung. The pulmonary parenchyma is disrupted by extensive areas of acute necrosis, edema, and hemorrhage. HE. (**E**) A higher magnification image demonstrates severe interstitial edema and effacement of faveolar walls by fibrin and necrotic debris admixed with abundant bacilli. (**F**) A Gram–Twort stain of the same section demonstrates abundant delicate, elongated Gram-negative bacteria.

## Data Availability

Not applicable.

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
