# Peer review of "Fatal Bronchopneumonia and Tracheitis in a Green Turtle (Chelonia mydas) Caused by Serratia proteamaculans"

_animals, 2022, doi:10.3390/ani12151891_

Round 1
Reviewer 1 Report
Consider these few minor changes and considerations for this manuscript:
Line 28: Change comprising fibrin to comprised of fibrin,
Line 32 (Keywords): remove clinical pathology and insert tracheitis
Line 53: Capitalize Twort
Line 80: delete multifocally
Line 81: Please consider adding a period after intestine and beginning the next sentence with These were
Figure 1B. This image is difficult interpret and the center is the case presentation. Can you please increase the magnification? Even if you need two or three images of this gross lesion, this will prove valuable.
Section 3.2 Histopathology: In the gross description, you talk about intestinal changes (lines 80 and 81). Please either describe the histo or remove those lines from the gross.
Discussion: This animal is reported by you to be in good weight. Can you please include a paragraph discussing the possibility of cold stunning setting this animal up for this infection?
Author Response
Dear Reviewer 1,
Thank you for your comments about this manuscript and for providing opportunities for improvement. We have reviewed your comments and provide the below responses:
Line 28: Change comprising fibrin to comprised of fibrin, DONE
Line 32 (Keywords): remove clinical pathology and insert tracheitis DONE
Line 53: Capitalize Twort DONE
Line 80: delete multifocally DONE
Line 81: Please consider adding a period after intestine and beginning the next sentence with These were DONE
Figure 1B. This image is difficult interpret and the center is the case presentation. Can you please increase the magnification? Even if you need two or three images of this gross lesion, this will prove valuable. An alternate image plate has been provided.
Section 3.2 Histopathology: In the gross description, you talk about intestinal changes (lines 80 and 81). Please either describe the histo or remove those lines from the gross. These lines have been removed from the gross description as they are not significant to this case.
Discussion: This animal is reported by you to be in good weight. Can you please include a paragraph discussing the possibility of cold stunning setting this animal up for this infection?
While “cold stunning” of sea turtles is routinely reported in North America and Europe, it is not a phenomenon of green turtles from New South Wales. In September 2020 the average water temperature around Sydney was 17-18oC (62.6-64.4oF). While it is important to consider environmental causes of debility, we feel that singling out cold stunning as a potential contributing factor in this case would be unjustified. However, in order to introduce this phenomena to the reader, 'cold stunning' has been introduced in the introduction as a recognised cause of respiratory disease in sea turtles, and a comment about water temperatures included in the case case description.
We trust that the changes made to the manuscript have strengthened the report, and look forward to seeing the manuscript progress.
Best regards,
J Hall, H Bender, N Miller, and P Thompson
Reviewer 2 Report
This is a well-composed and written case report of a novel pathogen in a sea turtle. While single case reports often do not have a hugely impactful contribution to the literature, the authors make a valid argument that this is a unique pathogen that has some more far-reaching implications as a cross-kingdom pathogen.
A few suggestions:
Introduction - I would recommend including a reference or two regarding cold-stunning, as this is an extremely common source of immunosuppression and opportunistic infection in sea turtles
Line 40: 'deceased' seems out of place, as the animal was alive initially on presentation
M&M: line 58 - a small amount of discussion might be warranted regarding why the cultures were incubated at 35C, as this is not a normal reptile temperature, and thus has the potential to falsely select for bacteria different that the primary pathogens in the living animal (I recognize this is standard for most microbiological laboratories, which is why it is important to note and discuss).
Results: line 77, 79: specify this image is Figure 1B
Author Response
Dear Reviewer 2,
Thank you for your comments about this manuscript and for providing opportunities for improvement. We have reviewed your comments and provide the below responses:
Introduction - I would recommend including a reference or two regarding cold-stunning, as this is an extremely common source of immunosuppression and opportunistic infection in sea turtles
Thank you. We recognise that cold-stunning is an important syndrome across North America and Europe and have added words to the introduction to reflect this. This turtle was found in the Australian spring when water temperatures around Sydney were, on average, 18oC and therefore cold-stunning is an unlikely contributor in this case. We have added additional words to reflect water temperatures so that this is more clear to the reader.
Line 40: 'deceased' seems out of place, as the animal was alive initially on presentation
The term deceased has been removed and instead the term fatal is used to reflect that the animal died rather than was found dead.
M&M: line 58 - a small amount of discussion might be warranted regarding why the cultures were incubated at 35C, as this is not a normal reptile temperature, and thus has the potential to falsely select for bacteria different that the primary pathogens in the living animal (I recognize this is standard for most microbiological laboratories, which is why it is important to note and discuss).
Thank you for raising this issue. We had neglected to include the normal reptile protocol of incubating HBA plates at room temperature in this manuscript. We revisited the worksheets on this case to ensure this protocol was followed and have placed additional information in the methods and results section to reflect this.
Results: line 77, 79: specify this image is Figure 1B
Figure 1 has been amended as per reviewer 1 comments. We trust that these changes, and additional changes to the text, are satisfactory in addressing this comment.
We trust that the changes made to the manuscript have strengthened the report, and look forward to seeing the manuscript progress.
Best regards,
J Hall, H Bender, N Miller, and P Thompson